# Improved Anther Culture Media for Enhanced Callus Formation and Plant Regeneration in Rice (*Oryza sativa* L.)

**DOI:** 10.3390/plants10050839

**Published:** 2021-04-22

**Authors:** Jauhar Ali, Katrina Leslie C. Nicolas, Shahana Akther, Azerkhsh Torabi, Ali Akbar Ebadi, Corinne M. Marfori-Nazarea, Anumalla Mahender

**Affiliations:** 1Rice Breeding Platform, International Rice Research Institute (IRRI), Los Baños, Laguna 4031, Philippines; ktrnlsl.nicolas@gmail.com (K.L.C.N.); C.Marfori-Nazarea@irri.org (C.M.M.-N.); m.anumalla@irri.org (A.M.); 2BRAC Agriculture and Research Center, 75 Mohakhali, Dakha 1212, Bangladesh; akthershahana88@gmail.com; 3Rice Research Institute of Iran, Agricultural Research, Education and Extension Organization (AREEO), Rasht 41996-13475, Iran; ebady_al@yahoo.com (A.T.); aliakbar.ebadi@gmail.com (A.A.E.)

**Keywords:** anther culture, media components, rice hybrids, callus induction, plant regeneration

## Abstract

Anther culture technique is the most viable and efficient method of producing homozygous doubled haploid plants within a short period. However, the practical application of this technology in rice improvement is still limited by various factors that influence culture efficiency. The present study was conducted to determine the effects of two improved anther culture media, Ali-1 (A1) and Ali-2 (A2), a modified N6 medium, to enhance the callus formation and plant regeneration of *japonica*, *indica*, and hybrids of *indica* and *japonica* cross. The current study demonstrated that genotype and media had a significant impact (*p <* 0.001) on both callus induction frequency and green plantlet regeneration efficiency. The use of the A1 and A2 medium significantly enhanced callus induction frequency of *japonica* rice type, Nipponbare, and the hybrids of *indica* × *japonica* cross (CXY6, CXY24, and Y2) but not the *indica* rice type, NSIC Rc480. However, the A1 medium is found superior to the N6 medium as it significantly improved the green plantlet regeneration efficiency of CXY6, CXY24, and Y2 by almost 36%, 118%, and 277%, respectively. Furthermore, it substantially reduced the albino plantlet regeneration of the induced callus in two hybrids (CXY6 and Y2). Therefore, the improved anther culture medium A1 can produce doubled haploid rice plants for *indica* × *japonica*, which can be useful in different breeding programs that will enable the speedy development of rice varieties for resource-poor farmers.

## 1. Introduction

Rice is one of the major staple food crops for more than 3.5 billion people who rely on this crop for almost 20% of their regular calorie intake (Ricepedia, http://ricepedia.org/rice-as-food/, accessed on 15 June 2020). Cultivars of the major subspecies *indica* are widely grown in Asian countries vis-à-vis *japonica* cultivars [1], and almost 89% of global rice production is consumed mainly in six Asian countries: China, India, Indonesia, Bangladesh, Vietnam, and Japan [2]. To feed the estimated 8.6 billion global population by 2030, 756 million metric tons (MMT) of annual rice production are required to provide food security [3]. Currently, rice is grown in more than 100 countries, with a total harvested area of approximately 158 million hectares, and approximately 470 million tons of produced milled rice (http://ricepedia.org/rice-as-food/rice-productivity/, accessed on 15 June 2020). Therefore, to ensure the food security of the rapidly growing global population, rice production must be increased to 852 million tons by 2035 [4]. The rice crop is affected by biotic and abiotic stresses that decrease yield productivity [5,6]. Thus, there is an urgent need to develop climate-smart rice cultivars with multiple biotic and abiotic stress tolerance through the use of advanced biotechnological tools and modern conventional breeding approaches.

To date, several high-yielding inbred and hybrid rice varieties have been developed through conventional and molecular breeding approaches [7,8,9,10,11,12,13]. However, the drastic changes in climatic environments such as temperature (cold and heat), water scarcity, flooding, salinity, nutrient deficiency in the soil, and increasing diseases and pests negatively affect grain yield [14,15,16,17,18]. Similarly, soil fertility and arable land availability have also decreased in rice-growing regions across the globe [19]. Therefore, to overcome these problems, it is necessary to identify and develop multiple-stress-tolerant rice varieties (MSTRV) and integrate them well into future breeding programs [13,20,21,22]. The advancement of genetic engineering technologies, modern conventional breeding, and in vitro anther culture (IAC) methods could be combined to develop MSTRV [23,24]. Although conventional breeding approaches have substantially enhanced cereal productivity, their progress rate is slow [25]. Thus, there is a need to respond quickly and make rapid changes to drastically reduce breeding cycles to sustain the crop under unpredictable environmental conditions. The application of IAC along with advanced biotechnological approaches could hasten the process to develop superior breeding materials or varieties by decreasing the number of breeding cycles dramatically. IAC is a promising technique that complements conventional breeding approaches to fast-track crop improvement [26]. For a long time, the IAC technique has been recognized as a valuable adjunct to breeding and has been used in the development of haploids/doubled haploids (DHs) in numerous crop species [27]. Two steps are mainly involved in IAC, involving the induction of organogenic calli from microspores and the other regeneration of green plantlets (RGP) from calli [28]. This has provided a strategy with many advantages, such as shortening the breeding cycle by fixation of homozygosity, broadening genetic diversity, increasing selection efficiency, allowing the early expression of recessive genes, and increasing genetic gain [29]. The plants derived from DH techniques are completely homozygous breeding lines that can be produced by anther or microspore culture within a year, instead of waiting for more than five generations of inbreeding cycles [30,31,32,33]. Moreover, theoretically, no further segregation can be expected from the developed DH plants, which makes them useful as a fixed homozygote mapping population for different molecular genetic studies. Each plant developed through IAC could be a potential homozygous line, which can be useful to study phenotypic variation for desirable traits [34,35]. The DH lines are also ideal for genetic mapping of agro-morphological and complex traits. Advantages of DHs are quickest homozygosity and uniformity. However, there are numerous drawbacks, including segregation ratio distortion, the incidence of albinism, a limited and frozen crossover, which significantly limits their application [24,26,28,32]. However, the potential of IAC in *indica* rice varieties is yet to be completely exploited due mainly to the recalcitrant genetic backgrounds in *indica* rice varieties [36]. Recently, Mishra and Rao [37] and Mayakaduwa and Silva [38] reviewed IAC applications in rice. Several researchers have been working to develop DHs in many different crops, including major cereals, economically imperative medicinal plants, trees, and fruit crops [39]. DHs’ production is a breakthrough to speed up rice varietal development [40], but several studies on the IAC technique have emphasized the importance of using different media components to optimizing the responses during callus formation and regeneration [31,32,37,41,42,43]. Compared with subspecies *japonica*, subspecies *indica* is still limited in anther response due to a few issues such as early necrosis, poor callus formation, and the increasing number of albino plantlets. Among the several factors, the genotype of the explants, growing conditions of the donor plants, media composition including macro-and micronutrients, vitamins, carbohydrates, organic adjutants, and growth regulators have been identified to influence the culture efficiency [44,45,46,47,48].

To increase the success rate of callus proliferation and regeneration, there is a need to overcome the gap between the key steps for determining the effective media composition and each component’s concentration in the IAC protocol. The compound colchicine that is widely used in IAC has also been shown to significantly improve green DH plant production in various crops [27,31,49]. The growth regulators that have been mostly used in IAC, such as 2,4-dichlorophenoxyacetic acid (2,4-D), naphthalene acetic acid (NAA) and cytokinin can be manipulated at different concentrations in the culture environment for improved callus induction (CI) and regeneration from anthers [50]. Further, the need to optimize basal media composition for callus growth has been well confirmed in obtaining improved callus proliferation and regeneration of green plantlets [46,47,51,52,53,54]. For instance, with a modification of MS medium composition, Sah et al. [55] achieved 82.66% of regeneration percentage in *japonica* rice cv. Kitaake. Similarly, Rout et al. [32] examined the effect of callus formation efficiency by using elite long-duration hybrid rice variety CRHR32. The N6 medium (2.0 mg L^−1^ 2,4-D + 0.5 mg L^−1^ BAP + 30 g L^−1^ maltose) showed the highest CIF, which is almost double that of MS medium [56]. Up to now, different types of media components have been used in several crops. Among the various types of growth media, the N6 medium has been widely used as a basal medium, and it has had an excellent response for CI [46]. Modified N6 medium, which contains 0 mg L^−1^ 2,4-D, 0.5 mg L^−1^ BAP, and 30 g L^−1^ maltose, is effective for callus formation vis-à-vis MS and SK1 media. According to Roy [57], the highest CI was obtained in aromatic *indica* rice varieties when the N6 medium was supplemented with 3% and 6% maltose. Similarly, another study by Gioi and Tuan [58] noted that the callus frequency in the LS medium was lower than with MS or N6 in *indica* × *japonica* hybrids. In modifying the growth media components, Zaidi et al. [51] found that maltose is the most effective carbohydrate as it contributed to the highest CIF and RGP. However, after further exploitation of anther culture in molecular breeding and genetic studies, many of the research efforts have been confirmed, mostly on the optimization of protocols for CI and RGP.

The anther culture response of subspecies *indica* on the basal N6 medium is quite restricted compared with the callusing and regeneration responses of *japonica* on the same basal medium [31,53,59]. Many other studies show that genotype and culture composition imposes a major influence on callusing and regeneration during IAC [32,33,41]. Therefore, it is important to investigate the effect of modified N6 medium composition on improving CIF and RGP in different rice subspecies.

This present study aims to improve anther culture efficiency in terms of callus formation and plant regeneration by using modified N6 [56] medium compositions, which were designated as Ali-1 (A1) and Ali-2 (A2) in five rice genotypes belonging to different subspecies. The major objectives included evaluating the anther response of the five selected rice genotypes to the modified N6 (A1 and A2) media compared with the widely used callus induction media such as L8 [60] and N6; calculating the RGP of three hybrids on modified A1 and A2 media; identifying the best-optimized media for CI and RGP.

## 2. Materials and Methods

### 2.1. Plant Materials

Five rice genotypes (origins), Nipponbare (NB) from the National Institute of Crop Science (Tsukuba, Japan), NSIC Rc480 (GSR8) from IRRI (Philippines), GSR-H-0014_CXYR-24 (CXYR 24) GSR-H-0017_CXY-6 (CXY6) (from the Chinese Academy of Agricultural Sciences (CAAS), Beijing, China), and Y-Liang You-2 (Y2) (from the Hunan Hybrid Rice Research Center (HHRRC), Changsha, China), were used in this experiment with proper material transfer agreements (MTA). Seeds of the five rice varieties were obtained from the Rice Breeding Platform, International Rice Research Institute (IRRI), Philippines (14.11° N, 121.15° E). The varieties belonged to different subspecies that have varied anther culture response. Nipponbare is a *japonica* genotype known to be highly responsive to anther culture; NSIC Rc480, on the other hand, belongs to subspecies *indica*, which is known to be a recalcitrant type; and GSR-H-0014_CXYR-24 GSR-H-0017_CXY-6 and Y-Liang You-2 are hybrids of *indica* and *japonica* crosses.

### 2.2. Selection of Panicles in Booting Stage and Cold Pre-Treatment

Healthy panicles under the booting stage of the five rice varieties were collected from the primary tillers of each genotype in the morning between 07:00, and 09:00 a.m. when the distance between the base of the flag leaf and the primary leaf node is 5–10 cm. The collected panicles were washed in running water and wiped with moistened cotton soaked in 70% ethanol. Surface-sterilized panicles in the booting stage were then wrapped in wet cheesecloth and aluminum foil and then sealed in polyethylene bags to prevent desiccation and to maintain pollen viability. Furthermore, the wrapped panicles were incubated at 8 °C for 7–8 days for cold pre-treatment.

### 2.3. Novel Media Compositions and Preparations

Stock solutions of the macronutrients, micronutrients, vitamins, phytohormones, and other chemical components of the four media tested were prepared. The required quantity of each chemical was dissolved in double-distilled water (DDW). The compositions of N6, L8, and modified N6 media (A1 and A2) components of each chemical concentration (addition or modified) were represented by an asterisk and hash symbols (Table 1). In the A1 medium, N6 was modified for its sugar source: sucrose was reduced to 30 g L^−1,^ and 30 g L^−1^ of maltose was added. Also, myo-inositol (a vitamin) was added at 100 mg L^−1^. Furthermore, plant hormones such as 2,4-D were reduced to 1 mg L^−1^, and 1 mg L^−1^ of NAA and 0.10 mg L^−1^ of zeatin were added. However, in A2 medium, N6 medium was modified by reducing the sucrose to 10 g L^−1^, adding 10 g L^−1^ of sorbitol sugar, increasing the maltose to 40 g L^−1^, and adding 10 mg L^−1^ of AgNO_3_, 1000 mg L^−1^ of yeast extract, 1 mg L^−1^ of NAA, 10 mg L^−1^ of glycine, and 0.10 mg L^−1^ of zeatin. The pH was adjusted to 5.8 in all four media before autoclaving at 121 °C for 20 min.

### 2.4. Anther Collection and Callus Induction Frequency

The panicles collected at the booting stage were dissected in a laminar flow cabinet (Figure 1a,b). Spikelets with anthers that occupy one-third to one-half of the total floret length were selected [61] (Figure 1c). This approximately corresponds to the late uni-nucleate and/or early bi-nucleate stage of pollen development, which is the most responsive CI stage [62]. The selected spikelets were soaked in 20% NaOCl solution for 20 min and rinsed four to five times with sterile distilled water. Anthers from the selected spikelets were isolated by cutting the base of the floret and tapping the tip on the side of the Petri dish (60 × 15 mm). A total of 60 anthers were inoculated in each Petri dish, which contained 10 mL of callus induction medium.

Four callus induction media were used: two modified N6 media (Ali-1 and Ali-2), L8 [60], and N6 [56]. Five replications were prepared for each treatment combination (media × genotype) with ten Petri dishes per replicate. The cultured anthers were incubated in 24-h dark conditions in a plant growth chamber for six weeks at a temperature of 25 °C with a relative humidity of 60–85% (Figure 1d). The anthers were examined for callus formation at weekly intervals (Figure 1e). The callus induction frequency of each genotype from the four callus induction media was recorded 6 weeks after inoculation. All cali (single or multiple) that originated from a single anther were all counted (Figure 1f,g). The callus induction frequency was recorded by following this formula: CIF = (Total number of calli induced/total number of cultured anthers) × 100.

### 2.5. Plant Regeneration

The calli induced from the four callus induction media that reached 2–3 mm in diameter were transferred to MS [63] regeneration medium supplemented with 1 mg L^−1^ 6-benzyl amino purine (BAP), 1 mg L^−1^ kinetin, and 1 mg L^−1^ α-naphthalene acetic acid (NAA), along with 0.1 g L^−1^ myo-inositol and 30 g L^−1^ sucrose, and solidified with 7 g L^−1^ phytagel after adjusting the pH to 5.8. Inoculated cultures were kept at 25 °C with a relative humidity of 60–85% and a 16-h photoperiod from white cool fluorescent tubes (Figure 1h,i). The regenerated green plantlets were then transferred to culture tubes that contained MS medium devoid of any phytohormones to promote root induction (Figure 1j–l). Completely regenerated plants with roots were acclimatized on a hydroponic nutrient solution [64] in greenhouse conditions (Figure 1m). Plantlets with well-developed roots were then planted in pots in greenhouse conditions for further observation and evaluation (Figure 1m–p). Multiple green plantlets derived from the same calli were individually planted and counted because their origin could be from different pollen grains, which could mean different recombinants. Data were gathered on the number of regenerated plants for the three rice genotypes at weekly intervals (Figure 1h), and the data on the green plantlet regeneration (RGP), albino plantlet regeneration (PAP) from induced callus, green plantlet regeneration efficiency (RGPE) and albino plantlet regeneration efficiency (PAPE) from 100 anthers were recorded and calculated as indicated here: RGP = (Total number of green plantlets regenerated/total number of induced callus) × 100; PAP = (Total number of albino plantlets regenerated/total number of induced callus) × 100; RGPE = (Total number of green plantlets regenerated/100 anthers) × 100; PAPE = (Total number of albino plantlets regenerated/100 anthers) × 100.

### 2.6. Data Analyses

The experiment was laid out as a two-factor factorial in a randomized complete block design (RCBD) wherein the variety was considered factor A and the medium as factor B with five replications for each treatment combination. Statistical analyses of all the data were performed in R software version 4.0.2 [65]. Analysis of variance (ANOVA) was performed for all the measured parameters (CIF, RGP, PAP, RGPE, PAPE), and significant differences between treatments were made by the Tukey HSD Test at the 5% level of significance.

## 3. Results

### 3.1. Callus Induction Frequency in Improved Media

The responses of five rice varieties to four different culture media for the frequency of CI and RGP were investigated. Anthers from five genotypes were cultured in N6, L8, A1, and A2 media. Three weeks after anther plating, visible calli were observed from individual anthers plated on the four tested callus induction media. The anthers were able to produce single or multiple calli. Morphologically, two types of calli were noticed: one was soft and translucent, which is considered non-embryogenic, while the other was white, compact, and embryogenic.

Analysis of variance revealed that medium and genotype had a highly significant effect on callus induction frequency (*p <* 0.001) (Table 2). Compared with *indica* rice variety NSIC Rc480, the hybrids from *indica* and *japonica* crosses, CXY-6, CXYR-24, and Y2, as well as Nipponbare showed a better CIF performance in the improved media (A1 and A2), as illustrated in Figure 2A. The highest CIF was observed in Y2 (50.17%) cultured in A2 medium, followed by CXY6 in A2 medium as well (49.60%). whereas the lowest CIF was observed in NSIC Rc480 (2.30%) in the L8 medium. The subspecies *indica* NSIC Rc480 showed poor CIF in the four tested media (Figure 2A). For the highly responsive genotype, Nipponbare, the use of the A1 medium increased the CIF by almost 158% and 62% compared to using the N6 and L8 medium, respectively. The use of the A2 medium also enhanced CIF by 145% and 55% compared to using the N6 and L8 medium, respectively. Even for the least responsive type (NSIC Rc480), enhanced callus formation was observed using the two improved callus induction media, but this was found not to be significant. Substantial improvement in the CIF was also observed among the tested hybrids using the A1 and A2 medium. The CIF potential of Y2, CXY6, and CXY24 was almost doubled compared to using the N6 medium (Figure 2A).

### 3.2. Effect on RGP by Using Improved Media

Induced calli that reached 2–3 mm in size were transferred to regeneration medium. After 2 weeks, it was observed that some of the transferred calli started producing green spots, which eventually differentiated into green shoots, whereas some of them produced white shoots that differentiated into albino plantlets. Data analysis on the three *indica* × *japonica* rice hybrids revealed that the callus induction medium greatly affects these genotypes’ green plantlet regeneration (RGP), albino plantlet regeneration (PAP) from induced callus as well as green/albino plant regeneration efficiency (RGPE/PAPE). The interaction between the genotypes and media was also found to be significant in all these measured parameters (Appendix A). The highest RGP was observed from initiated calli in the A1 medium, which ranged from 72.11% to 117.48%, followed by the N6 medium (42.01% to 141.33%), the L8 medium (13.94% to 87.83%), and the lowest in the A2 medium (13.84% to 64.39%) (Figure 2B). The A1 medium doubled the RGP potential of Y2 (from 60.32% to 117.48%) and CXY24 (from 42.01% to 72.11%) compared to the N6 medium, except for the CXY6 hybrid (from 141.33% to 108.77%). Furthermore, the A1 and A2 medium showed significantly reduced PAP of the CXY6 and Y2 hybrids compared to the N6 and the L8 medium except for the CXY24 (Figure 2C). The highest PAP was recorded in the L8 medium (6.70% to 79.95%) while the lowest PAP was recorded in the A2 medium (16.42% to 30.05%) (Figure 2C). Across the three hybrids tested, the highest RGPE was recorded in the A1 medium (20.43% to 57.17%) while the lowest RGPE were in the L8 medium (1.80% to 4.40%) (Figure 3A). Furthermore, the use of the A1 medium instead of the N6 medium significantly improved the RGPE of the CXY6, the CXY24, and the Y2 hybrids by 36%, 118%, and 277%, respectively (Figure 3A). Also, the A1 medium significantly reduced the PAPE of the Y2 hybrid by almost 14% compared to using the N6 medium (Figure 3B).

## 4. Discussion

Improvement of inbred rice varieties through conventional breeding approaches is a tedious process since it requires at least 8 to 10 generations of inbreeding to produce the desired level of homozygosity, and it is not usually conceivable to reach 100% homozygosity [66]. Therefore, to overcome this, the IAC technique is a promising approach to develop doubled haploids in numerous crop species [26]. The major benefit of IAC is the existence of spontaneous chromosome doubling, which results in the creation of homozygous DHs in a single generation, and these are the most fertile and cytologically stable lines, except that a few of them exhibit chromosomal abnormalities [39,67,68]. This technique also has several advantages in selection efficiency and improving genetic gain [29]. However, the success rate of IAC depends on several factors such as genotypic effects, selection of panicles, type of culture medium, seasonal effects, cold pre-treatment, the combination of growth hormones, the concentration of micro-and macronutrients, and culture conditions [69,70,71].

To date, more than 280 varieties in 33 plant species of different crops have been developed through DH technology [39,66,69,72]. These include crops such as barley (*Hordeum vulgare* L.), maize (*Zea mays* L.), rice (*Oryza sativa* L.), pepper (*Capsicum annuum* L.), rapeseed (*Brassica napus* L.), tobacco (*Nicotiana tabacum* L.), and wheat (*Triticum aestivum* L.). Further, to enhance the efficiency of the IAC technique, several authors, such as Gioi and Tuan [54], Trejo-Tapia et al. [50], Zaidi et al. [51], and Rout et al. [32] followed different strategies through modification of plant growth hormones and different combinations of media and their concentrations. However, compared with *japonica*, *indica,* and *indica* × *japonica* hybrids of rice varieties have not so far achieved a significant increase for CI and RGP. Therefore, in the current study, the androgenic response of five rice genotypes with varied anther culture response was tested on two improved N6 media, A1 and A2, in comparison with the two most commonly used anther culture media, N6 and L8. Several authors attested to the superiority of N6 [53,62,73,74,75] and L8 media [76,77] in promoting androgenesis in rice.

### 4.1. Effect of Improved Media on Callus Induction

The performance of the tested genotypes in this study was also consistent with the general trend of variation in anther culture response wherein *japonica* rice has a favorable response compared with *indica* × *japonica* and *indica*, and hybrids formed more calli than *indica* rice (*japonica* > *indica* × *japonica* > *indica*). The A2 medium was improved by changing the concentration of macronutrients, micronutrients, and vitamins; supplementing it with silver nitrate, myo-inositol, and yeast extract; and using a combination of different types and concentrations of carbohydrate sources (sucrose, maltose, and sorbitol) and phytohormones (2,4-D, NAA, zeatin). In the A1 medium, the modifications of the N6 medium involved only using an additional carbohydrate source (maltose), changing the concentration of sucrose, incorporating a different type of phytohormones, and adding myo-inositol to the medium. Several authors reported that the ratio of nitrate ion and ammonium ion in a culture medium could dictate the success of anther culture [37,38,47].

A high level of KNO_3_ and (NH_4_)SO_4_ in a culture medium such as N6 was reported to be effective for *japonica* rice types, but not for *indica* [47,78]. The results of this study supported this observation, wherein higher CIF was observed in Nipponbare and the hybrids, which have a *japonica* background, while the lowest CIF in all tested media was found in the NSIC Rc480, which has an *indica* background. The addition of myo-inositol, the use of different types and concentrations of carbohydrate sources, and the growth regulators could also be another reason for the enhanced CIF observed in the genotypes, including the *indica* rice line cultured in A1 and A2 media. Silver nitrate, an ethylene inhibitor, has been reported to enhance CI in rice [53,59,73]. It prevents the early senescence of cultured anthers by inhibiting ethylene biosynthesis, which is possibly beneficial in IAC [47]. The use of silver nitrate was effective not only in rice but also in Brassicas and cotton [79,80,81,82]. However, the use of an appropriate concentration of silver nitrate is essential to evaluate the action of ions that may be dependent on the available concentration in the medium [83]. Moreover, the time of culture and the appropriate concentration of Ag+ also affect the number of green plants obtained [84].

Myo-inositol, although not essential for maintaining culture viability, was also reported to significantly enhance in vitro response in monocots. Yeast extract, an organic adjutant, was also reported to enhance CI in rice [57,85]. It was found to be important in enhancing callus initiation in *indica* genotypes. The type and concentration of the carbohydrate sources in anther culture media were also found to affect callus production in rice significantly. These sources not only provide energy but also maintain osmotic pressure in the culture environment. Sucrose had been the primary carbon source in many anther culture media, but the beneficial effect of maltose in enhancing anther culture response has been well documented [28,37,48,75,86,87]. The advantage of maltose over sucrose might be due to the relatively slow decomposition rate of maltose in culture medium than sucrose, which allows it to act as an osmoticum that stabilizes the culture medium, thus preventing plasmolysis of the microspores [78,88]. The use of maltose was also reported as a promoter of CI not only in rice but also in other cereals such as wheat, barley, and rye [38,89].

Many of the developed anther culture media such as N6 [56], B5 [90], and SKI [52] used only one type of carbon source, but the improved media employed a combination of sucrose and maltose. Just like maltose, sucrose also acts as an osmoticum that maintains the culture’s osmotic potential, enabling the nutrients to be readily available for absorption of the explant for optimum proliferation [91]. On the other hand, sorbitol was also another carbon source added to the A2 medium. It promoted the production of embryogenic calli in maize that has a better regenerative capacity [92]. The beneficial effects of the different carbohydrate sources reported by previous researchers were the reasons for using maltose, sucrose, sorbitol, sucrose, and maltose in A1 and A2 media, respectively. However, the promotive effects on CI of maltose and other carbohydrate sources decline beyond the optimum requirement of genotypes [87]. Shahnewaz and Bari [93] had found a significant effect of sucrose concentration on the frequency of callus induction in rice, and the highest CI and RGP were observed for BRRI dhan29 on medium supplemented with 4% sucrose

Other major differences of the improved media from L8 and N6 were the type, ratio, and concentrations of phytohormones used in those media. Phytohormones played a significant role in the androgenic responses of rice. It specifically influenced the callusing ability of the anthers [94]. The type and concentration of these phytohormones in culture media, especially auxins and cytokinins, significantly affect the callusing ability and regeneration potential of the induced calli. 2,4-D and NAA are the most commonly used auxins for anther culture, while kinetin, zeatin, and BAP are the most commonly used cytokinins [95]. Manipulating the ratio of auxins and cytokinins profoundly affects the CI and plant regeneration potential of different rice genotypes. For this study, the ratio and concentrations of the phytohormones for L8 and N6 were based on the published literature, and this was different from the concentration of A1 and A2 media (Table 1). The L8, A1, and A2 media used both auxin and cytokinin, but L8 used a much higher concentration of auxin (0.5 mg L^−1^ 2,4-D and 3.5 mg L^−1^ NAA) than the improved media (1 mg L^−1^ NAA and 1 mg L^−1^ 2,4-D); N6, on the other hand, did not use any cytokinin and used only 2 mg L^−1^ 2,4-D. Enhanced CI was reported on a medium that uses a combination of auxin and cytokinin (2,4-D, NAA, and kinetin; 2,4-D and kinetin; NAA and kinetin) at a ratio with higher auxin than cytokinin [74,88,96]. The results of this study supported the previous researchers’ observation on the improved CI from the media that used combined auxin and cytokinin at a ratio with higher auxin. However, this observation was not observed on the L8 medium, which also has a higher auxin and cytokinin ratio. The poor response of the tested genotypes on this medium might be attributed to the higher doses of auxin sources (0.5 mg L^−1^ 2,4-D, 3.5 mg L^−1^ NAA), which might be detrimental to callus formation.

### 4.2. Effect of Improved Media on Plant Regeneration

Across all the hybrids tested, the results revealed that the use of A1 medium as a callus induction medium might have a positive effect eventually on the green plant regeneration efficiency of the genotypes. The type of sugar in the anther culture medium was reported to affect the differentiation of pollen calli on regeneration media [97]. The use of maltose in the callus induction media together with sucrose as the carbohydrate source might be one of the factors that could have enhanced the ability of the calli induced from these media to regenerate into green plants. Lentini et al. [73] claimed that maltose enhanced green plant formation in rice. It was also found better suited for direct plantlet regeneration, especially when NAA was added to the callus induction medium [78]. However, Bagheri et al. [98] argued that sugar source and concentration do not significantly affect plant regeneration, but highly affect only CI; as a result, the authors suggested using sucrose for plant regeneration because it is cheaper, but using maltose for callus induction. The poor green plant regeneration of calli induced from the L8 medium may be attributed to higher auxin concentration. This study supported several authors’ observations on the reduced ability of calli induced from anther culture medium that used a higher concentration of auxins [47,52] to regenerate green plantlets.

A total of 7116 green plantlets were produced from the calli of the three rice hybrids induced from the four tested callus induction media. Calli induced from the A1 medium were able to regenerate a total of 3797 green plantlets while calli from the A2 medium were able to regenerate a total of 1249 green plantlets, 1819 in N6 and only 254 in the L8 medium. The production of a higher number of green plantlets is a requisite for anther culture techniques to be incorporated into any breeding program. However, this is hindered by the production of more albino plantlets in many anther culture experiments. Albinism remains a major challenge that limits the application of this technology in many breeding programs, and the *indica* rice type was more prone to this problem [99,100,101]. Kumari et al. [100] believed that this trait is controlled by one or two genes with low heritability. However, factors such as pre-treatment duration and the use of sucrose beyond optimum concentration have been reported to promote albino production in rice. Prolonged cold pre-treatment of anthers beyond the optimum was shown to significantly increase albino incidence [78]. Chen [102] reported the highest numbers of albino plants regenerated from calli induced from media containing 9% sucrose. Data analysis from this study revealed an increase in the albino plantlet regeneration efficiency (PAPE) in the A1 medium for CXY24 but not in CXY6 and Y2. The high efficiency of albino production observed on N6, A1, and A2 might be due to the genotype and media composition. The N6 medium for this study used 6% sucrose, while A1, A2, and L8 used only 1%, 3%, and 5% sucrose, respectively. However, the estimated 2–5% of sucrose levels were good for anther culture in rice [103]. The results of this study highlighted the significance of the genotype, and media compositions and their interactions on CIF and RGP efficiency (RGPE) in rice. The modified N6 media (A1) can be useful to continue studying the anther response, CIF, and RGPE in the IAC technique in rice.

## 5. Conclusions

Anther culture technique is a biotechnology tool that has found importance and a niche in expediting many crop breeding processes. However, in rice this technology was highly limited to *japonica* rice types because of the recalcitrant nature of many *indica* rice types. In this study, two improved anther culture media were evaluated based on their ability to enhance the callus formation and green plant regeneration of five rice genotypes that belonged to different subspecies, which have varied anther culture responses. The results indicated that the media and genotype significantly affect callus formation as well as green plant regeneration efficiency. The use of A1 media significantly enhanced callus induction potential, as well as efficiency of green plantlet regeneration as compared to the N6 and the L8 media. Thus, A1 as callus induction media for anther culture of rice can produce doubled haploid rice plants, which could be beneficial in many rice breeding programs. However, in the future, the prospects for IAC and the integration of genomics tools provide novel opportunities for improving selection efficiency, maximizing genetic grain, and improving varietal development, leading to the early release of rice varieties with desirable traits.

## Figures and Tables

**Figure 1 plants-10-00839-f001:**
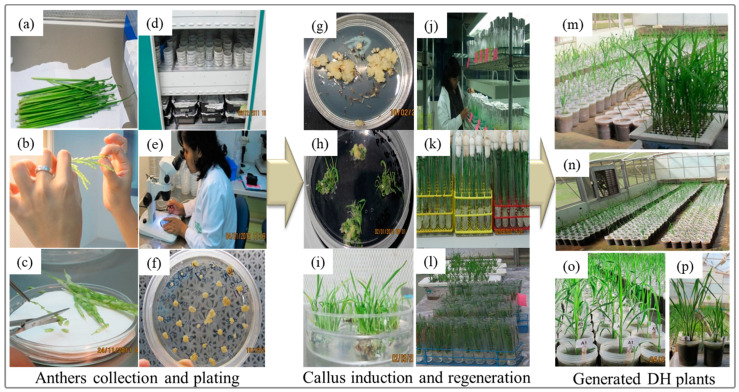
Depicted diagram of the generation of DH plants. (**a**) Boot collections; (**b**) Selection of spikelets at the proper stage; (**c**) Cutting the base of the floret and anther inoculation process; (**d**) Plated anther incubated at 25 °C for 6–7 weeks for callus induction; (**e**) Examination of the anther plates at weekly interval for anther response and callus formation; (**f**) Calculation of callus induction frequency; (**g**) Callus proliferation; (**h**,**i**) Green plantlet regeneration in the regeneration medium; (**j**) Vigorous growing green plantlets in the test tube containing MS medium and incubated at 30 °C under 24 h light; (**k**,**l**) Regenerated green plantlets transferred to hydroponic nutrient solution and grown in the greenhouse at 29–30 °C temperature for two weeks; (**m**–**o**) Individual plants originating from single callus transplanted in pot condition; (**p**) Difference between haploid (right) and diploid (left) rice plants.

**Figure 2 plants-10-00839-f002:**
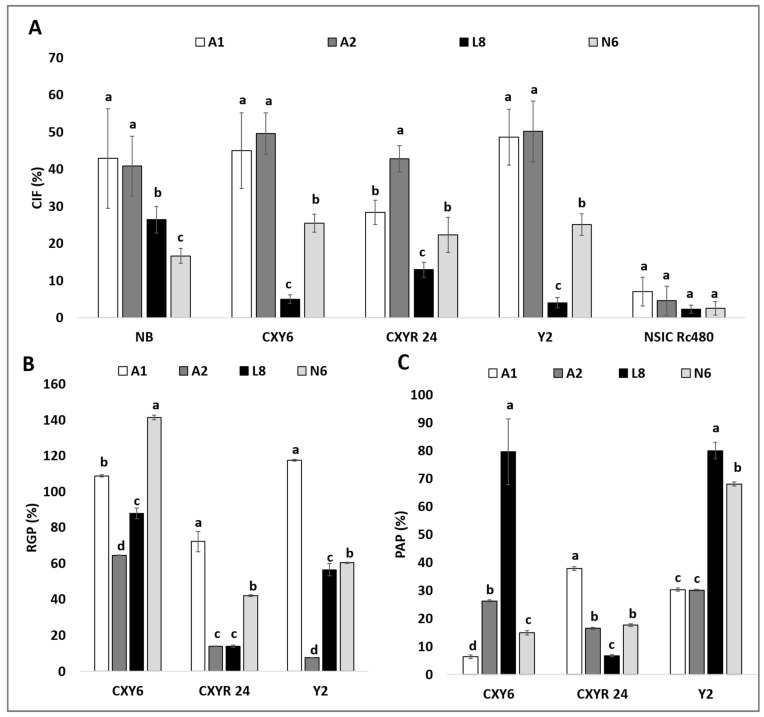
Callus induction frequency (CIF), green plantlet regeneration (RGP), and albino plantlet regeneration (PAP) of different genotypes in the tested media. (**A**) Comparison of callus induction frequency (CIF) of different genotypes incubated in the test media. (**B**) Comparison of green plantlet regeneration (RGP) of hybrids of *indica* × *japonica* incubated in the test media. (**C**) Comparison of the albino plantlet regeneration (PAP) of hybrids of *indica* × *japonica* incubated in the test media. The data presented are means of five replicates, and error bars represent SD. Values with different letter (a,b,c,d) per genotype are significantly different (Tukey HSD Test, *p* < 0.05, *n* = 5 replicates).

**Figure 3 plants-10-00839-f003:**
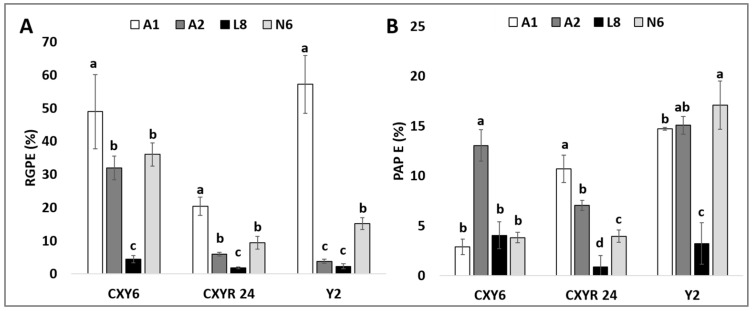
Plant regeneration efficiency of the different hybrids incubated in the test media. (**A**) Comparison of the green plantlet regeneration efficiency (RGPE) of 100 anthers from the hybrids of *indica* × *japonica* incubated on the tested media. (**B**) Comparison of the albino plantlet regeneration efficiency (PAPE) of the 100 anthers from the hybrids of *indica* × *japonica* incubated on the test media. The data presented are means of five replicates, and error bars represent SD. Values with different letter (a,b,c,d) per genotype are significantly different (Tukey HSD Test, *p* < 0.05, *n* = 5 replicates).

**Table 1 plants-10-00839-t001:** Nutrient composition of four callus induction media used for the study.

Chemicals	Concentration (mg L^−1^)			
	L8	N6	Ali-1 (A1) (Modified N6)	Ali-2 (A2) (Modified N6)
Macronutrients				
MgSO_4_·7H_2_O	185.00	185.00	185.00	200.00 *
KH_2_PO_4_	540.00	400.00	400.00	500.00 *
KNO_3_	3000.00	2830.00	2830.00	3000.00 *
CaCl_2_·2H_2_O	150.00	166.00	166.00	150.00 *
(NH_4_)_2_SO_4_	-	463.00	463.00	300.00 *
Micronutrients				
H_3_BO_3_	6.00	1.60	1.60	2.00 *
MnSO_4_·4H_2_O	17.98	4.40	4.40	5.00 *
ZnSO_4_·7H_2_O	10.00	1.85	1.50 *	3.00 *
Na_2_MoO_4_·2H_2_O	0.250	-	-	0.250 ^#^
CuSO_4_·5H_2_O	0.025	-	-	0.025 ^#^
CoCl_2_·6H_2_O	0.025	-	-	0.025 ^#^
KI	0.83	0.80	0.80	0.83 *
FeSO_4_·7H_2_O	27.85	27.80	27.80	43.00 *
Na_2_EDTA·2H_2_O	37.25	37.30	37.30	56.00 *
AgNO_3_	-	-	-	10.00 ^#^
Vitamins				
Nicotinic acid	3.00	0.50	0.50	2.00 *
Thiamine HCl	2.50	1.00	1.00	4.00 *
Pyridoxine HCl	5.00	0.50	0.50	2.00
Myo-inositol	100.00	-	100.00 ^#^	100.00 ^#^
Others				
Sucrose	50,000.00	60,000.00	30,000.00 *	10,000.00 *
Sorbitol		-	-	10,000.00 ^#^
Maltose	-	-	30,000.00 ^#^	40,000.00 ^#^
Glycine	-	10.00	10.00	10.00
Phytagel	8000.00	2000.00	2000.00	2000.00
YE	-	-	-	1000.00 ^#^
Lactalbumin hydrolysate	300.00	-	-	-
Phytohormones				
2,4-D	0.50	2.00	1.00 *	1.00 *
NAA	3.50	-	1.00 ^#^	1.00 ^#^
Kinetin	2.00		-	-
Zeatin	-	-	0.10 ^#^	0.10 ^#^
pH	5.80	5.80	5.80	5.80

* Chemical modified from N6 media; ^#^ Addition to N6 media components.

**Table 2 plants-10-00839-t002:** Two-way ANOVA for effects of media and genotype on callus induction.

S. no	Sources of Variation	Degrees of Freedom	Sum of Squares	Mean Square
1	Genotype	4	11,443	2861 ***
2	Media	3	12,779	4260 ***
3	Genotype x media	12	5395	450 ***
4	Residuals	80	2497	31

*** Significant at the 0.001 probability level.

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
