# Peer review of "Improved Anther Culture Media for Enhanced Callus Formation and Plant Regeneration in Rice (Oryza sativa L.)"

_plants, 2021, doi:10.3390/plants10050839_

Round 1

Reviewer 1 Report

Dear Authors,

The manuscript is fairly well written; however, I do not see any significant novelty in it. There have been several recent articles on very similar topics.  In addition, it has the following deficiencies:

It lacks photographic documentation.

No information about the origin of the tested lines. Do they come from different crosses or are they from a single one?

It is not clear how many anthers in vitro culture was initiated from. Was it 60 anthers x 5 repetitions x 4 media or 12 anthers x 5 repetitions x 4 media? If the latter then the results may be biased by too few explants.

If replicates were made then why is there no SD for each genotype x medium configuration in Table 3? This raises the suspicion that the variability within g x m configuration  was greater than between them.

Table 4 is missing results for the other two genotypes

The problem is obtaining high regeneration efficiencies from in vitro cultures of indica rice anthers. Analyses in which a single genotype is used contribute very little to the development of this field. A more numerous group of genotypes must be used for progress to occur.

Best regards

Reviewer 2 Report

The manuscript entitled “Improved anther culture medium for enhanced callus formation and plant regeneration in rice (Oryza sativa L.)” described the effect of 4 different induction culture media on callus production of one japonica and one indica genotypes and three hybrids of indica and japonica and also on green and albino plant regeneration of the 3 hybrids.

The recalcitrance of indica cultivars is well-documented. Therefore, modifications of protocols that enhance the efficiency of DH plant production specially from indica cvs.,  but also from hybrids (indica x japonica and japonica x indica), have a great interest for plant breeding.

The manuscript can be accepted for publication after major revision.

From my point of view authors should provide more data to increase the value of the manuscript:

The effect of the four induction medium on the efficiency of green DH plant production should be presented in the 2 cvs. and the 3 hybrids. Thus besides the variable percentage of anther producing callus (% callus induction),  the authors should also show the total number of green and albino plants/total number of callus (RGP and PAP, respectively),  the number of green plants/100 anthers and also the percentage of spontaneous doubling (or the number of green DH plants/100 anthers) in the five genotypes. In this way, it will be easier to see which medium A1 or A2 produces the highest efficiency.

Furthermore, since ANOVA for callus induction shows a significant interaction genotype x treatment, means separation of the 4 culture media for each genotype should be presented (Table 3). ANOVA for RGP, PAP and the new variables requested should also be performed (at least for the number of green plants/100 anthers,) and means separation of the different media for each genotype should be presented in case there is a statistically significant interaction genotype x medium.

It seems that A1 medium is more efficient than A2 (both media produce similar rates of callus induction but the percentage of green plant regeneration is higher with A1 medium) for the 3 hybrids. The main differences between N6 and A1 media are the growth regulators composition and the carbohydrate source. I suggest performing a new experiment to know which of these components is responsible for the enhanced efficiency or if they have an additive effect.

Concerning discussion, the authors should be very cautious about the results and discussion of the effect of A2 on induction and regeneration. A2 produces similar rates of green plants than N6 only in CXY-6 but not in the other 2 hybrids. Many compounds have been changed in A2 medium compared with N6 and L8, so you cannot really know which of them are responsible for the increased value of these variables. The discussion should be focused also on the effect of the media on the variable number of green plants/100 anthers, which reflects the efficiency of the system. Also,  authors should point out which is the best medium A1 or A2.

Minor corrections:

I understand that the induction medium is solid; otherwise different calli coming from the same anther should be identified and individualized, instead of considering all of them as a single one callus. Please mention the solidified agent used in the culture media.

The concentration of zeatin mentioned in the table 1 and text is different. Please clarify.

Lane 250, add, respectively, after higher than N6 and L8 medium.

Line 254-255:  mention also the high CIF observed with A2 on CXR-24 hybrid (49.60%).  

Line 284:  add that A2 medium produced a lower RGP than N6 medium

Round 2

Reviewer 1 Report

Dear Authors,

Although I am not a fan of this type of paper, I find the manuscript acceptable in its current form.

Regards,

Reviewer

Author Response

No comments to be addressed and we thank the reviewer.

Reviewer 2 Report

The manuscript has been improved. However, there are still some aspects that need to be clarified.

A clear definition of RGP and PAP should be provided. Are these variables: Total number of green and albino plants/total number of anther inoculated)*100 that are  Number green and albino plants/100 anthers or not? If this is so, the author should be conscious that the variable number of green plants/100 anthers defines the efficiency of the system, thus special emphasis should be given in Results and Discussion to this variable. 

More modifications should be performed in the manuscript, which are indicated in the attached file. 

Author Response

A clear definition of RGP and PAP should be provided. Are these variables: Total number of green and albino plants/total number of anther inoculated)*100 that are  Number green and albino plants/100 anthers or not? If this is so, the author should be conscious that the variable number of green plants/100 anthers defines the efficiency of the system, thus special emphasis should be given in Results and Discussion to this variable. 

Clear definitions of RGP and PAP have now been provided. We have now improved the Results and Discussion sections accordingly in the revised MS

Round 3

Reviewer 2 Report

The quality of the presentation has been improved. Please see comments on the manuscript. English revision of the new text incorporated in the manuscript needs to be done. Also, review the name of the variables RGP and PAP through the manuscript.

Author Response

We have now carried out the required corrections in the MS. We also reviewed the name of the variables RGP and PAP through the manuscript.